# Alpha Enolase 1 Ubiquitination and Degradation Mediated by *Ehrlichia chaffeensis* TRP120 Disrupts Glycolytic Flux and Promotes Infection

**DOI:** 10.3390/pathogens10080962

**Published:** 2021-07-30

**Authors:** Bing Zhu, Jere W. McBride

**Affiliations:** 1Department of Pathology, University of Texas Medical Branch, Galveston, TX 77555-0609, USA; bizhu@utmb.edu; 2Department of Microbiology & Immunology, University of Texas Medical Branch, Galveston, TX 77555-0609, USA; 3Center for Biodefense and Emerging Infectious Diseases, University of Texas Medical Branch, Galveston, TX 77555-0609, USA; 4Sealy Center for Vaccine Development, University of Texas Medical Branch, Galveston, TX 77555-0609, USA; 5Institute for Human Infections and Immunity, University of Texas Medical Branch, Galveston, TX 77555-0609, USA

**Keywords:** *Ehrlichia*, tandem repeat protein, effector, ubiquitin, HECT ligase, α-enolase

## Abstract

*Ehrlichia chaffeensis* modulates numerous host cell processes, including gene transcription to promote infection of the mononuclear phagocyte. Modulation of these host cell processes is directed through *E. chaffeensis* effectors, including TRP120. We previously reported that TRP120 moonlights as a HECT E3 Ub ligase that ubiquitinates host cell transcription and fate regulators (PCGF5 and FBW7) to promote infection. In this study, we identified a novel TRP120 substrate and examined the relationship between TRP120 and α-enolase (ENO1), a metalloenzyme that catalyzes glycolytic pathway substrate dehydration. Immunofluorescence microscopy and coimmunoprecipitation demonstrated interaction between ENO1 and TRP120, and ubiquitination of ENO-1 by TRP120 was detected in vivo and in vitro. Further, ENO-1 degradation was observed during infection and was inhibited by the proteasomal inhibitor bortezomib. A direct role of TRP120 Ub ligase activity in ENO-1 degradation was demonstrated and confirmed by ectopic expression of TRP120 HECT Ub ligase catalytic site mutant. siRNA knockdown of ENO-1 coincided with increased *E. chaffeensis* infection and ENO-1 knockdown disrupted glycolytic flux by decreasing the levels of pyruvate and lactate that may contribute to changes in host cell metabolism that promote infection. In addition, we elucidated a functional role of TRP120 auto-ubiquitination as an activating event that facilitates the recruitment of the UbcH5 E2 ubiquitin-conjugating enzyme. This investigation further expands the repertoire of TRP120 substrates and extends the potential role of TRP120 Ub ligase in infection to include metabolic reprogramming.

## 1. Introduction

*Ehrlichia chaffeensis* is a Gram-negative, obligately intracellular bacterium and etiologic agent of human monocytotropic ehrlichiosis (HME), an emerging, life-threatening, tick-borne zoonosis [1,2]. *E. chaffeensis* preferentially infects mononuclear phagocytes and survives intracellularly by subverting innate immune defenses mediated in part by tandem repeat protein (TRP) effectors [3]. TRPs are secreted intracellularly during infection by the type-1 secretion system, where they interact in a spatial and context-dependent manner with a diverse array of host cell proteins involved in essential cellular processes [4,5,6,7,8,9]. Most recently, *E. chaffeensis* TRP120 effector was shown to act as a ligand mimic that activates Wnt and Notch signaling pathways, as well as possessing other moonlighting functions during infection [10,11,12,13,14]. 

TRP120 has been observed in the host cell cytosol and nucleus during infection, where it interacts with an array of host proteins including many involved in post-translational modifications, such as enzymes required for ligation and conjugation of ubiquitin (Ub) and ubiquitin-like modifier (SUMO) [4,9]. TRP120 is conjugated with SUMO at a canonical consensus motif that enhances interactions with defined host protein targets, and facilitates recruitment of these proteins to the ehrlichial vacuole [12]. More recently, we reported that TRP120 moonlights as a HECT-type E3 ubiquitin (Ub) ligase that ubiquitinates host proteins for proteasomal degradation [15]. Inhibiting HECT Ub ligase activity significantly decreased TRP120 interactions and recruitment of host proteins to ehrlichial inclusions and negatively impacted *E. chaffeensis* infection, demonstrating an important role during infection [15]. Additionally, we have shown that TRP120 directly targets FBW7 for ubiquitination, resulting in FBW7 degradation by the ubiquitin–proteasome pathway to maintain the stability of Notch and other oncoproteins involved in cell proliferation and apoptosis regulation [16].

Alpha-enolase (ENO-1) is an important glycolytic enzyme which is found in the cytoplasm, nucleus and surface of eukaryotic cells. ENO-1 catalyzes the dehydration of 2-phosphoglycerate to phosphoenolpyruvate in the catabolic glycolytic pathway, which is responsible for the ATP-generated conversion of 2-phosphoglycerate to phosphoenolpyruvate during glycolysis through substrate-level phosphorylation [17,18,19,20,21]. ENO-1 has been found to play other roles in inflammation, tumor suppression and monocyte and mast cell differentiation [20,21,22,23]. Disruption of glycolysis through the reduced expression of ENO-1 has been linked to cellular stress, bacterial and fungal infections and cancer, as well as autophagic and catabolic pathway adaptations [24,25]. Although the significance of the TRP120-ENO-1 interaction during infection is unknown, we recently demonstrated that knockdown of ENO1 significantly enhanced *E. chaffeensis* infection [6], a finding that suggests that ENO-1 may be degraded to promote infection.

Covalent conjugation to ubiquitin is a major post-translational modification that regulates protein stability, function and localization. Ubiquitylation is achieved by the sequential actions of a ubiquitin-activating enzyme (E1), a ubiquitin-conjugating enzyme (E2) and an E3. The specificity and efficiency of ubiquitination are largely determined by the E3 [26]. There are two major types of E3 ligases: the RING E3s that function as scaffolds to transfer E2-Ub into the E3 and substrate. For HECT E3s, a major step for the formation of a polyubiquitin chain is the recruitment of E2~Ub to the E3 cystine catalytic site, resulting in the formation of an E3~Ub intermediate and transfer to the substrate. In previous studies, it has been shown that auto-ubiquitination of Mdm2, a RING E3, leads to strong recruitment of E2-conjugating enzymes, overcoming the rate-limiting step of E2 recruitment and increasing the processivity of ubiquitination [27]. We recently reported that TRP120 directly targets PCGF5 and FBW7 for ubiquitination in the presence of UbcH5b/c [15]. However, the molecular mechanism of Ub ligation and Ub chains formation of TRP120, and the functional roles in host cell E2/E3 Ub enzyme interactions remain undefined. 

In this study, we demonstrate that during *E. chaffeensis* infection, TRP120 directly targets ENO-1 for ubiquitination and degradation via the ubiquitin–proteasome pathway. ENO-1 degradation slowed the rate of production of pyruvate and lactate, thereby disrupting host glucose metabolism. Further, we show that *E. chaffeensis* TRP120 is auto-ubiquitinated with K48 Ub-linkage chains and auto-ubiquitination of TRP120 leads to recruitment of E2-conjugating enzymes that increase the processivity and ubiquitination of TRP120 substrates. 

## 2. Results

### 2.1. TRP120 Directly Interacts with ENO-1

We previously demonstrated that knockdown of TRP120-interacting host proteins including ENO-1 significantly increased infection [6], suggesting that this protein may be degraded to promote infection by TRP120. We first examined ENO-1 levels during infection and found a significant decrease in ENO-1 beginning at 48 h post infection (hpi) (Figure 1A). Coimmunoprecipitation (Co-IP) was performed to determine the interaction between TRP120 and ENO-1, and direct interaction between ENO-1 and TRP120 was detected in infected THP-1 cells (Figure 1B). To further explore this interaction, we performed confocal immunofluorescent microscopy on uninfected and *E. chaffeensis-*infected THP-1 cells at 24, 48 and 72 hpi. In uninfected and *E. chaffeensis-*infected THP-1 cells, ENO-1 was mainly found in cytoplasm, but strong colocalization between ENO-1 and TRP120 expressing *E. chaffeensis* morulae was observed (Figure 1C). The fluorescent intensity of the ENO-1-TRP120 colocalization was quantified using image J (Figure 1D), illustrating temporal changes in localization of ENO-1 with TRP120-expressing *E. chaffeensis* morulae. There was a reduction in ENO-1 levels throughout infection from 24 to 72 hpi, while TRP120 levels increased as the infection progressed. At 72 hpi, ENO-1 cytoplasm presence was reduced (~4 fold) compared to uninfected cells. These results demonstrate that TRP120 and ENO-1 interact, and ENO-1 levels decrease during *E. chaffeensis* infection.

### 2.2. TRP120 Targets ENO-1 for Ubiquitination and Proteasomal Degradation

In order to show that ENO-1 degradation occurs via the ubiquitin proteasome pathway by TRP120, we examined ENO-1 ubiquitination levels during infection. *E. chaffeensis*-infected cell lysates were subjected to ubiquitin enrichment, and a novel ENO-1 band (~60 kD) was identified in infected cells, but not in uninfected cells (Figure 2A), demonstrating that ENO-1 is ubiquitinated during infection. TRP120 has HECT E3 ligase activity and two host cell substrates (PCGF5 and FBW7) have been identified [15,16]. Hence, based on this initial finding, we considered the possibility that ENO-1 is a TRP120 substrate. To address this question, we examined the ubiquitination of ENO-1 in vitro in the presence of recombinant TRP120. Immunoblot analysis of products with anti-ENO-1 and anti-Ub demonstrated the presence of a higher-molecular-mass species of ENO-1 (60 kD) in the presence of TRP120, whereas this band was not detected in the control (Figure 2B, demonstrating that TRP120 directly ubiquitinates ENO-1. To further analyze the role of TRP120 in ENO-1 stability, we transfected TRP120-WT and TRP120-C520S, a mutant TRP120 lacking the E3 ligase function in HeLa cells, and examined ENO-1 stability. Indeed, there was a progressive reduction in ENO-1 levels compared to controls in cells transfected with TRP120-WT and ENO-1 levels were stable in cells expressing TRP120-C520S (Figure 3A). 

The 26S proteasome is an essential protease complex responsible for the degradation of ubiquitinated proteins. To elucidate a mechanism of TRP120-mediated ENO-1 degradation through the ubiquitin–proteasome pathway, we further examined the ENO-1 levels in bortezomib-treated and untreated cells. Decreased ENO-1 was observed at 24, 48 and 72 hpi in untreated cells, but remained unchanged in bortezomib-treated cells (Figure 3B,C). These studies indicate that TRP120 directly targets ENO-1 for ubiquitination and ENO-1 appears to be degraded by the ubiquitin–proteasome pathway. 

### 2.3. ENO-1 Knockdown Promotes E. chaffeensis Infection

In vitro and cellular analysis of ENO-1 ubiquitination demonstrated that ENO-1 is a novel substrate of TRP120 E3 ligase activity. To further examine the effects of ENO-1 levels on infection, THP-1 cells were treated with ENO-1 siRNA and infected with *E. chaffeensis*. Significant knockdown of ENO-1 by siRNA was observed by immunoblot analysis (Figure 4A). Moreover, THP-1 cells transfected with ENO-1 siRNA had significantly more ehrlichial morulae/cell at 24, 48 and 72 hpi (Figure 4B) than cells treated with control siRNA, indicating that the level of ENO-1 has a significant impact on ehrlichial infection (Figure 4C). The infection in ENO-1 knockdown and control cells was also assessed by qPCR, and a significant increase in bacterial load was detected at 24, 48 and 72 hpi (Figure 4D–F). These findings support the conclusion that ubiquitination and degradation of ENO-1 promote *E. chaffeensis* infection.

### 2.4. ENO-1 Knockdown Alters Host Cell Metabolism

Pyruvate, the end product of glycolysis in the presence of ENO-1, plays a major role in cell metabolism. In the presence of the cytosolic enzyme lactate dehydrogenase, pyruvate is further converted to lactate. Both pyruvate and lactate are keystone molecules critical for numerous aspects of eukaryotic and human metabolism. Thus, we examined the potential functional role of ENO1 degradation in disrupting glycolytic flux to promote infection. We measured the intracellular production and secretion of pyruvate and lactate in uninfected and *E. chaffeensis-*infected cells and found that the secretion of pyruvate and lactate was significantly (*p* ≤ 0.05) reduced at 24, 48 and 72 hpi compared to uninfected cells (Figure 5A,C). Moreover, the intracellular pyruvate and lactate production was also significantly (*p* ≤ 0.05) decreased during infection (Figure 5B,D). To further demonstrate that the decrease in pyruvate and lactate production was due to the degradation of ENO-1, we compared changes in pyruvate and lactate production before and after ENO-1 knockdown. After silencing of ENO-1, the extracellular and intracellular levels of pyruvate and lactate were decreased in uninfected cells (Appendix A). In addition, the levels of pyruvate and lactate were significantly decreased after ENO-1 knockdown during infection (Appendix A). To determine if other metabolic enzymes, such as pyruvate dehydrogenase, which is involved in the conversion of pyruvate to lactate, we investigated pyruvate dehydrogenase levels and found that they were similar in uninfected and infected cells (Appendix A). These results suggest that ENO-1 is an essential enzyme molecule that regulates pyruvate and lactate production during infection, and elimination of ENO-1 slows the rate of production of pyruvate and lactate. These results also suggest that the degradation of ENO-1 by TRP120 may be associated with metabolic reprogramming of host cells to promote infection.

### 2.5. TRP120 Is Auto-Ubiquitinated with K48-Linked Poly Ubiquitin

Our laboratory previously reported that TRP120 is post-translationally modified by ubiquitin (Ub) and that autoubiquitination occurred in vitro in the presence of host UbcH5b/c E2 enzymes through intrinsic and host-mediated HECT ligase activity. TRP120 ubiquitination sites were also mapped and confirmed by ectopic expression of TRP120 lysine mutants in cells [15]. However, the ubiquitin chain types have not been fully characterized. Thus, we characterized TRP120 ubiquitin chains and focused on ubiquitin chains branched at K48 and K63, because K48 and K63 chains are the two most abundant and functionally significant linkages. Mutation of lysine 63 or 48 to arginine mutants renders ubiquitin (Ub) unable to form poly-ubiquitin chains via lysine 63 and 48 linkages with other ubiquitin molecules. To identify ubiquitinated chains of TRP120, the K48R and K63R ubiquitin mutants were used to identify the lysines being utilized for ubiquitin chain linkage formation. We observed a strong band at 150 kDa with Ub-WT and Ub-K63R by Western immunoblot using anti TRP120, anti-FK2 and anti-K48 antibodies that was consistent in mass with bands detected in the TRP120 auto-ubiquitination immunoblot [15]. However, mutation of K48R resulted in reduced TRP120-Ub level (Figure 6A), demonstrating that TRP120 was able to specifically autocatalyze and form K48-linked polyubiquitination chains in the presence of UbcH5b in vitro. We then examined *E. chaffeensis* TRP120 ubiquitin chains during infection by immunoprecipitating TRP120 from *E. chaffeensis*-infected cells and detected a band that reacted with anti-K48, but not K63 antibody by Western blot (Figure 6B). This finding is consistent with results obtained using the in vitro ubiquitination assay with K48R and K63R ubiquitin mutants.

### 2.6. Auto-Ubiquitination Facilitates the Interaction of TRP120 with the UbcH5 E2 Enzyme

The conserved catalytic cysteine residue in the C-terminus of known HECT E3 ligases has an important role in mediating E2 specificity and catalysis [28]. Because UbcH5b is required for TRP120 autoubiquitination [15], we considered the possibility that UbcH5b may interact with both unmodified and ubiquitinated TRP120. To test this hypothesis, we first examined UbcH5b protein levels and the interaction between TRP120 and Ubch5b in *E. chaffeensis*-infected host cells. Immunoblot analysis demonstrated that UbcH5b protein levels were significantly increased at 24, 48 and 72 hpi (Figure 7A). Immunofluorescence microscopy also revealed that UbcH5b colocalized with TRP120-expressing *E. chaffeensis* morulae (Figure 7B). To further demonstrate the interaction between TRP120 and UbcH5, coimmunoprecipitation was performed using *E. chaffeensis*-infected whole THP-1 lysates and anti-TRP120 and UbcH5b antibodies. As shown in Figure 7C, UbcH5b coimmunoprecipitated with TRP120, suggesting that these proteins interact in infected cells. Next, we investigated the mechanisms by which auto-ubiquitination of TRP120 facilitates the recruitment of UbcH5b E2 and stimulates TRP120 E3 activity. An in vitro pulldown assay was performed using GST UbcH5b, and ubiquitinated and unmodified TRP120. The eluted samples were subjected to a immunoblot assay using TRP120 antibody. As shown in Figure 7D, strong binding was observed between UbcH5b and unmodified TRP120. Meanwhile, strong auto-ubiquitinated TRP120 binding to its specific UbcH5b E2 enzyme was also observed, suggesting that specific recruitment of the TRP120–Ub–UbcH5b complex increases TRP120 E3 activity.

## 3. Discussion

Our recent investigations revealed that *E. chaffeensis* infection is dependent on the exploitation of host PTM pathways. This includes the demonstration that *E. chaffeensis* TRP120 moonlights as an HECT-type E3 ubiquitin (Ub) ligase that ubiquitinates host proteins for proteasomal degradation [15]. We previously showed that the transcription repressor, polycomb group ring finger protein 5 (PCGF5), is a substrate of TRP120 Ub ligase activity, resulting in PCGF5 degradation and enhanced infection [15]. Moreover, we showed that TRP120 directly targets FBW7 for ubiquitination and results in FBW7 degradation to maintain the stability of Notch and other oncoproteins involved in cell survival and apoptosis [16]. In this study, we identified a novel TRP120 substrate and demonstrated that ENO-1 is a substrate of TRP120 Ub ligase activity, resulting in ENO-1 degradation, which promotes infection.

ENO-1 is a glycolytic enzyme which has been found to play other roles in inflammation, tumor suppression and monocyte and mast cell differentiation [17,18]. To understand the functional roles of ENO-1 during infection, we first determined its cellular distribution, as this had been shown to play a role in its function. We determined that ENO-1 was mainly located in the cytoplasm, and that this would be related to its primary role in glycolysis, the conversion of 2-phosphoglycerate to phosphoenolpyruvate [17]. Strong colocalization was observed with confocal microscopy between TRP120 and ENO-1 primarily in cytoplasm during infection, which is consistent with the previously observed temporal/spatial dynamics of TRP120 cytoplasm accumulation and function [29]. 

Under cellular conditions, ENO-1 regulation occurs at both transcriptional and post-translational levels [25]. Substantial effort has been made to understand the regulatory mechanisms of ENO-1 activity [25]. Nuclear factor kappa B (NFkB) and hypoxia-inducible factor1α(HIF-1α) are involved in the transcriptional activation of ENO-1 [30,31]. MiR-206 may target ENO-1 and inhibit the expression of ENO-1 [32]. Activation of the Src and MEK/ERK signal pathways upregulates ENO-1 expression [33]. Further studies have shown that FBXW7 (FBW7) is a novel regulator of ENO-1 protein. FBXW7 physically interacted with and facilitated the ubiquitination and degradation of ENO-1 [34]. Consequently, the biological activity of ENO-1 was inhibited by FBXW7 [34]. However, in the context of *E. chaffeensis* infection of monocytes, we determined that FBW7 is progressively degraded and the level of FBW7 is not restored at any time during infection despite upregulated FBW7 gene expression [16]. This suggests that during infection FBW7 may not facilitate ubiquitination of ENO-1; hence, TRP120 is likely to play a critical role for ENO-1 ubiquitination to regulate levels during infection. 

The TRP120-ENO1 interaction revealed that TRP120-mediated ubiquitination-dependent degradation of ENO-1 is critical for *E. chaffeensis* infection. We previously showed that the reductions in ENO-1 through iRNA knockdown lead to enhanced ehrlichial infection. In this study, we determined that the degradation of ENO-1 by TRP120 E3 ligase activity is a mechanism that results in reduced levels of ENO-1. However, the cellular consequences of ENO-1 degradation during infection are not known. We observed that *E. chaffeensis* infection caused significant decreases in pyruvate and lactate production. This finding is consistent with the role of ENO-1 in dehydrating 2-phosphoglycerate to phosphoenolpyruvate, then to pyruvate, which is then converted to lactate in the presence of pyruvate dehydrogenase [17]. Pyruvate dehydrogenase was essentially indistinguishable in uninfected and infected ENO-1 knockdown cells, which suggests that TRP120 directly affects lactate production by mediating the degradation of ENO-1. Moreover, this finding highlights the fact that the rates of production of pyruvate and lactate, which are major products of glycolysis, are reduced, suggesting that TRP120 regulates the host cell metabolism by degrading ENO-1.

Many bacteria and viruses are known to hijack cellular metabolism for infection [35,36,37]. They actively remodel the host cell metabolism to redirect glycolysis and mitochondrial TCA cycle intermediates towards the biosynthesis of the lipid droplets, fatty acids, amino acids and nucleotides that are required for pathogen nutritional and survival needs [38]. On the other hand, host cell metabolic switches and pathways control the duration and intensity of innate or adaptive immune activation and further enhance infection [39]. In either case, the end result is generally a reduction in the tricaboxylic acid (TCA) cycle and an induction of aerobic glycolysis (also called the Warburg effect), in which pyruvate generated from glycolysis is converted to lactate in the cytoplasm, generating two ATP molecules for every glucose molecule. [40]. It has been shown that disruption of the TCA cycle increases the ability of *Salmonella* Typhimurium to survive within resting and activated murine macrophages [41]. Macrophages infected with *M. tuberculosis* exhibit reduced levels of TCA cycle and induce a similar shift towards aerobic glycolysis. Inhibition of this metabolic switch leads to reduced production of cytokine and enhanced growth of intracellular bacteria [36]. The present study shows that both pyruvate and lactate levels are significantly reduced during infection and suggests that the degradation of ENO-1 by TRP120 may have a direct role in pyruvate reduction, resulting in reduced TCA cycle and amino acid consumption and altered mitochondrial localization. On the other hand, it is not clear if *E. chaffeensis* infection induces aerobic glycolysis during infection for its replication or inhibits this metabolic switch to dampen the host response (i.e., production of cytokines). Therefore, future research will further elucidate the role of TRP120-mediated ubiquitination and degradation of ENO-1 in promoting *E. chaffeensis* infection. In addition, under normal conditions in most cells, pyruvate is shuttled into mitochondria, where it is oxidized via the TCA cycle, eventually generating ATP with which to promote cell growth [42]. Conversely, once inside mitochondria, mitochondria LDH would catalyze the conversion of lactate back to pyruvate, which would be oxidized through the PDH reaction to acetylCoA to induce cell growth and proliferation [43,44]. Therefore, it is less likely that downregulation of ENO-1 has a direct role in lactate reduction. However, whether ENO-1 downregulation by TRP120 has a direct role in lactate production through an induction of aerobic glycolysis is less clear. Therefore, our future research will elucidate if *E. chaffeensis* infection may induce aerobic glycolysis and further demonstrate that ENO-1 downregulation by TRP120 has a direct role or not in lactate reduction.

Polyubiquitin chains of different topologies regulate diverse cellular processes. K48- and k63-linked chains, the two most abundant chain types, regulate proteolytic and signaling pathways, respectively [45]. The K48-linked polyubiquitin chain was first characterized as a proteasome delivery signal for short-lived cytosolic proteins [46,47]. Proteasome-dependent protein degradation is classically mediated by covalent modification of the Lys-48-linked polyubiquitin chain. TRP120 undergoes auto-ubiquitination in the presence of UbcH5b/c with K48-Ub. It was therefore surprising that K48-specific ubiquitination of TRP120 in vitro is not associated with proteasomal degradation of TRP120 during infection. Indeed, that the K48-specific ubiquitination of TRP120 during infection was barely detectable suggests that the other ubiquitin linkage-specific ubiquitination of TRP120 may exist in the cellular context. One possible explanation for this phenomenon could be that TRP120 is ubiquitinated by host E3 ligases and the other ubiquitin linkages may exist and may have partial functions to maintain the stability of TRP120 during infection. We previously found that NEDD4L levels steadily increased during infection and demonstrated that NEDD4L strongly interacts with TRP120 and facilitates TRP120 ubiquitination in vitro [15]. NEDD4L activity has been associated with chain formation and substrate ubiquitination mainly via Lys-11, Lys-63, Lys-6, Lys-27 and Lys-29 linkages [48,49]. This suggests that other ubiquitin linkages may exist to play an important role in the regulation of TRP120 stability during infection that are difficult to detect. We previously reported that TRP120 ubiquitination can only be detected in enriched ubiquitinated lysates using specific TRP120 antibodies during infection due to the low levels of TRP120 ubiquitin conjugates present during infection [15]. 

In the ubiquitination system, ubiquitin-conjugating enzymes (E2s) are the central players in the trio of enzymes responsible for the attachment of ubiquitin (Ub) to E3s and substrate proteins [50]. The HECT families of structurally distinct E3 ligases use a catalytic cysteine residue to accept activated Ub in the form of an E2∼Ub thioester adduct; the E3 itself then catalyzes the aminolysis reaction to form an isopeptide bond between Ub and the acceptor lysine substrate [51]. In contrast, the RING-related E3s lack a catalytic cysteine and instead stimulate the ability of the E2 itself to catalyze aminolysis between the E2-conjugated Ub and the acceptor lysine substrate through an allosteric mechanism [52]. The multidomain RING-containing protein Mdm2 is the principal ubiquitin ligase of the pre-eminent tumor suppressor p53 [27]. In previous studies, the ubiquitinated Mdm2, but not unmodified Mdm2, was shown to bind strongly with UbcH5c. The fact that polyUb chains on Mdm2 enhance the recruitment of E2 enzymes indicates that autoubiquitination of Msm2 can overcome the rate-limiting step of E2 recruitment and increase the processivity of ubiquitination [27]. We previously demonstrated that TRP120 has a HECT-like E3 ligase domain and Cys^520^ is critical for TRP120 E3 Ub ligase activity and autoubiquitination in the presence of UbcH5b/c [15]. In this study, an in vitro pulldown assay showed that both unubiquitinated and ubiquitinated TRP120 were strongly bound to UbcH5b, suggesting that UbcH5b interacts with the active site of cysteine in both unubiquitinated and autoubiquitinated TRP120. In addition, it is well known that UbcH5b also catalyzes Lys48 chain topologies when acting with the E6-Ap HECT E3 ligase [53]. Our results show that TRP120 conjugates to polyubiquitin K48 chains and the ubiquitination of TRP120 occurs at five lysine residues (K33, K120, K130 K396 and K432) suggest that polyubiquitin chains on these five lysine residues (K33, K120, K130 K396 and K432) of TRP120 may enhance the recruitment of UbcH5 enzymes [15]. 

It is well known that ENO-1 has glycolytic and non-glycolytic functions and participates in a high number of cellular processes, suggesting that ENO-1-targeted therapeutic approaches may be considered. Future studies to clarify the functions of ENO-1 and its metabolites during infection will help us to better understand how such therapies could be developed. 

In this study, we demonstrated that *E. chaffeensis* TRP120 is essential for redirecting cellular metabolism through ENO-1 degradation by the ubiquitin–proteasome pathway during infection. In addition, the current study demonstrates that TRP120 self-ubiquitination occurs mainly through the K48 chain and reveals that auto-ubiquitination of TRP120 is not responsible for TRP120 degradation in vivo. We show that auto-ubiquitination of TRP120 leads to recruitment of E2-conjugating enzymes and increases the processivity of ubiquitination. Our findings illustrate the evolving knowledge regarding functional roles of TRP120 intrinsic and host HECT ligase activity in host–pathogen interactions. Future research will further elucidate the exact molecular mechanisms of alterations of mitochondrial localization, the impact of THP-1 cell replication and metabolic substrates, states and pathways in response to infection after ENO-1 knockdown and help us to further understand how degradation of ENO-1 modulates host cellular metabolism to promote infection. 

## 4. Materials and Methods

### 4.1. Cell Culture and Cultivation of E. chaffeensis

Human monocytic leukemia cells (THP-1) were grown in RPMI medium 1640 with L-glutamine and 25 mM HEPES buffer (Invitrogen, Carlsbad, CA, USA) supplemented with 1 mM sodium pyruvate (Sigma, St. Louis, MO, USA), 2.5 g/liter D-(+)-glucose (Sigma) and 10% fetal bovine serum (HyClone, Logan, UT, USA). *E. chaffeensis* (Arkansas strain) was cultivated in THP-1 cells as previously described [54].

### 4.2. Cell Lysis and Protein Extraction

*E. chaffeensis-*infected THP-1 cells were harvested at 24, 48 or 72 h post infection (hpi) and whole-cell lysates were extracted using a whole Cell Extraction Kit (Abcam, Cambridge, MA, USA). Plasmids carrying the full length of TRP120 wild type and TRP120^C520S^ mutant were transfected into HeLa cells with 4.0 to 8.0 ug of endotoxin-free plasmid purified with a plasmid maxikit (Qiagen, Valencia, CA, USA) using Lipofectamine 2000 (Invitrogen, Carlsbad, CA, USA) according to the manufacturer’s direction. Transfected HeLa cells were lysed using a Whole Cell Extraction Kit (Abcam, Cambridge, MA, USA). The concentration was determined by Lowry method (DC assay; BioRad, Hercules, CA, USA). For the Western blot experiment in Figure 3B, 26S proteasome inhibitor, bortezomib (Thermo Fisher Scientific, Waltham, MA, USA), was added to cell culture at 10 ng/mL concentration for 10 h before whole cell lysates were collected. 

### 4.3. Recombinant TRP120, ENO-1, Antibodies, and siRNAs

Generation of the TRP120 wild type and TRP120^C520S^ mutant expression construct (pAcGFP1 and pBAD/Thio) and expression and purification of recombinant TRP120 have been described previously [12,15,29]. ENO-1 recombinant protein was obtained from a commercial source (Abnova, Taipei, Taiwan). Rabbit or mouse anti-TRP120 antibodies have been described previously. Other antibodies used in this study were anti-FK2 (Enzo), anti-K48 (CST), anti-K63 (CST), anti-UbcH5b (CST), anti-ENO-1 (Abcam), anti-PDH (Santa Cruz) and anti-GAPDH (Proteintech, Rosemont, IL, USA). All antibodies used for immunofluorescence were tested by the vendor to ensure specificity and confirmed by Western immunoblotting, immunofluorescent microscopy or both. ENO-1 (siRNAs) and a control siRNA were obtained from GE Dharmacon (Lafayette, CO, USA). 

### 4.4. Western Blot Analyses

Protein samples were resolved by SDS-PAGE, transferred to nitrocellulose membrane and blocked in Tris-buffered saline with 5% non-fat dry milk. Primary antibodies used for Western blot assay were rabbit anti-TRP120 (1:5000), rabbit anti-UbcH5b (1:1000), rabbit anti-ENO-1 (1:1000), anti-PDH (1:1000) and rabbit anti-GAPDH (1:4000). Secondary antibodies were horseradish peroxidase-labeled goat anti-rabbit IgG and goat anti-mouse IgG (Kirkegaard & Perry, Gaithersburg, MA, USA). Densitometric quantification of the immunoblot bands was performed using ImageJ densitometry software.

### 4.5. Immunofluorescence Microscopy and Co-Immunoprecipitation

Uninfected or *E. chaffeensis*-infected THP-1 cells were cytocentrifuged onto glass slides, fixed in ice-cold3% paraformaldehyde in phosphate-buffered saline (PBS) for 20 min then permeabilized and blocked with 0.3% Triton X-100 and 2% bovine serum albumin in PBS for 1 h. The cells were incubated with primary antibodies for 1 h, which included rabbit anti-UbcH5b (1:100) or rabbit anti-ENO-1 (1:100) and mouse anti-TRP120 (1:1000). Samples werewashed and stained with fluorescein isothiocyanate (FITC)-conjugated or Alexa Fluor 594-IgG(H + L) and Alexa Fluor 488-IgG(H + L) secondary antibodies (1:100; Molecular Probes) for 30 min. Slides were washed then mounted with ProLong Gold Antifade reagent with DAPI (4′,6-diamidino-2-phenylindole) (Invitrogen). Images were obtained using an Olympus BX61epifluorescence microscope and were analyzed using Slidebook software (ver. 5.0; Intelligent Imaging Innovations, Denver, CO, USA). Coimmunoprecipitation was performed on highly infected (95%) THP-1 cells 3 d.p.i. using the EZ-Magna ChIP A/G immunoprecipitation kit (EMD Millipore, Billerica, MA, USA) with anti-*E. chaffeensis* TRP120 and IgG control antibodies and immunoblotting was performed with anti-ENO-1 antibody according to the manufacturer’s instructions.

### 4.6. In Vitro Ubiquitination and Binding Assays

TRP120 auto-ubiquitination assays were performed using an in vitro ubiquitination kit (Enzo Life Sciences, Farmingdale, NY, USA) with recombinant TRP120, ubiquitin wild type (Ub-WT), UB-k48R and Ub-K63R. ENO-1 ubiquitination was performed with recombinant TRP120, ENO-1. TRP120 (10 nM) and ENO-1 (100 nM) were added to ubiquitination buffer in the presence of E1 and UbcH5c and Ub-WT. Ubiquitination reactions were performed at 37 °C for 4 h, andthe reactions were stopped by addition of Laemmli buffer. Samples were boiled for 5 min and resolved by 4–20% SDS-PAGE for Western blotting using anti-TRP120, anti-ENO-1 and anti-Ub antibodies. *E. chaffeensis*-infected cell lysates (400 µg) were subjected to ubiquitin pulldowns (Enzo Life Sciences, Farmingdale. NY, USA) using Ubiquitin Enrichment Kit (Pierce, Appleton, WI, USA) following the manufacturer’s protocol and eluates were analyzed by Western blotting using anti-TRP120 and anti-ENO-1 antibodies. 

The auto-ubiquitinated TRP120 or unmodified TRP120 were prepared, respectively in the presence of UbcH5b E2s with or without ATP using an in vitro ubiquitination kit (Enzo Life Sciences, Farmingdale, NY, USA). For TRP120 and UbcH5b binding, we added the recombinant GST-UbcH5b protein to the prepared glutathione sepharose medium and incubated for at least 30 min at room temperature, using gentle agitation, such as end-over-end rotation. The medium was sedimented by centrifugation at 500× *g* for 5 min. the The supernatant was carefully decanted and the glutathione sepharose medium washed by adding 1 mL of PBS, repeated three times, and the auto-ubiquitinated TRP120 or unmodified TRP120 were added to Sepharose medium and incubated for 2 h. The glutathione sepharose was washed three times with PBS buffer, and the bound proteins were eluted by PBS with 5 mM reduced glutathione and analyzed by Western blotting.

### 4.7. RNA Interference and Real-Time qPCR

Specific ENO-1 siRNA was siGENOME SMARTpool siRNA (Dharmacon, Lafayette, CO). The control siRNA was ON-TARGET *plus* non-targeting siRNA (Dharmacon, Lafayette, CO). THP-1 cells (1 × 10^6^/well on a 6-well plate) were transfected with human ENO-1 siRNA (5 pmol) and control siRNA(5 pmol) using Lipofectamine 3000 (Invitrogen, Carlsbad, CA, USA). The cells were synchronously infected by cell-free *E. chaffeensis* at an MOI of ~50 after 1 day post transfectionand cell extractions were performed according to the manufacturer’s protocol at 24, 48 or 72 hpi. Diff-Quik-stained slides were collected and analyzed with an Olympus BX61 epifluorescence microscope using a color camera. Lysates were prepared using a whole cell extract kit (Abcam, Cambridge, MA, USA ) at room temperature and analyzed by Western blot. Total DNA was extracted from *E. chaffeensis*-infected THP-1 cells with a DNeasy Blood and Tissue Kit (Qiagen, Valencia, CA, USA) and bacterial load was measured by real-time quantitative PCR (qPCR). Amplification of the integral ehrlichial gene *dsb* was performed using Brilliant II SybrGreen master mix (Agilent, Santa Clara, CA, USA), 200 nM forward primer (5′-GCTGCTCCACCAATAAATGTATCCCT-3′) and 200 nM reverse primer (5′-GTTTCATTAGCCAAGAATTCCGACACT-3′). The absolute *E. chaffeensis dsb* copy number in the cells was determined against the standard curve as previously described.

### 4.8. Pyruvate and Lactate Assay

Supernatant and cell lysates from uninfected, *E. chaffeensis-*infected cells, ENO-1 siRNA knockdown and control (scrambled) siRNA cells then infected with *E. chaffeensis* were diluted with assay buffer, and the pyruvate and lactate assays were carried out following the manufacturers’ protocol (BioVision, Milpitas, CA, USA). Final measurements of pyruvate concentration were carried out at 570 nm on a microplate reader and measurements of lactate concentration were carried out at Ex/Em = 535/590 nm in a microplate reader.

### 4.9. Statistical Analysis

Statistical analyses were performed with a two-tailed unpaired *t* test. *p* < 0.05 was considered statistically significant.

## Figures and Tables

**Figure 1 pathogens-10-00962-f001:**
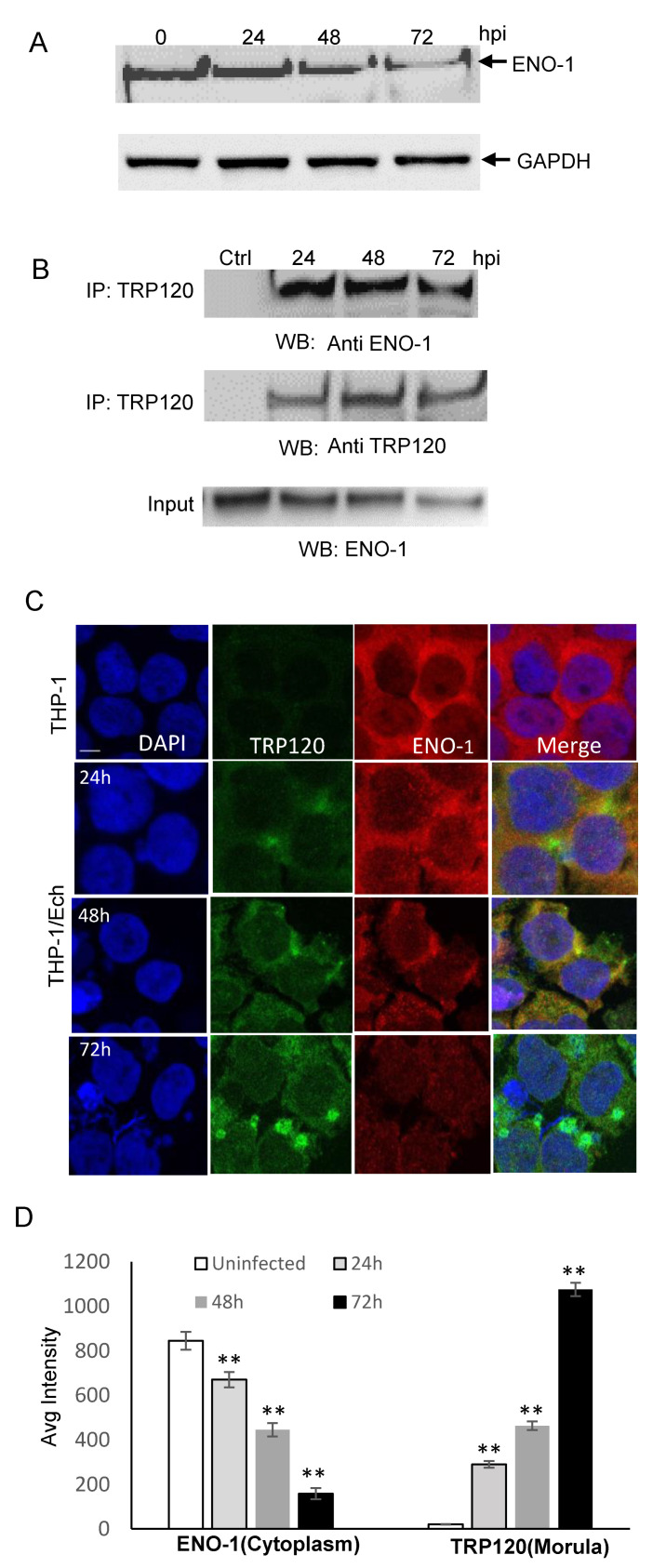
ENO-1 colocalizes with *E. chaffeensis* inclusions and interacts with TRP120. (**A**) Western immunoblot analysis of ENO-1 levels during *E. chaffeensis* infection at 1, 6, 24, 48 and 72 hpi. (**B**) Co-immunoprecipitation was performed to demonstrate direct interaction between TRP120 and ENO-1. THP-1 cells were infected with cell-free *E. chaffeensis* and the harvested at 24, 48 and 72 hpi. Western immunoblot detection of TRP120 and ENO-1 interaction using TRP120- and ENO-1-specific antibodies. (**C**) Immunofluorescence microscopy visualization shows ENO-1 colocalizing with TRP120-expressing *E. chaffeensis* morulae. THP-1 cells were probed with DAPI (DNA; blue), anti-TRP120 (green) and anti-ENO-1 (red) (top panel). *E. chaffeensis-*infected THP-1 cells were probed (24, 48 and 72 hpi) with DAPI (DNA; blue), anti-TRP120 (green) and anti-ENO-1 (red) (bottom panel). Bar, 10 µm. (**D**) ENO-1 and TRP120 fluorescent intensities from the confocal immunofluorescent microscopy (**A**) were calculated using image J and graphically demonstrated to be significant (Student’s test; **, *p* < 0.01). ENO-1 levels were reduced at 24, 48 and 72 hpi in cytoplasm compared to uninfected cells, and a significant increase in TRP120 expression on morulae was found.

**Figure 2 pathogens-10-00962-f002:**
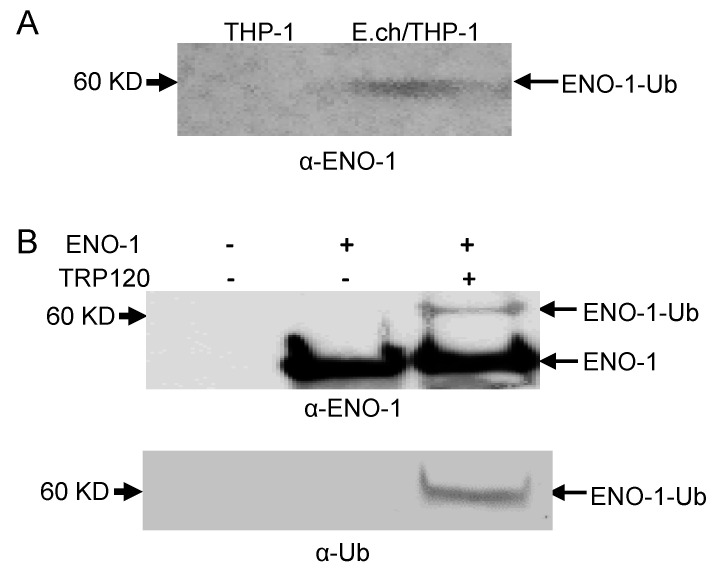
Ubiquitination of native and recombinant TRP120. (**A**) Uninfected and *E. chaffeensis-*infected THP-1 cell lysates were subjected to Ub enrichment and ubiquitinated ENO-1 was detected by Western blot. (**B**) In vitro ENO-1 ubiquitination in the presence of TRP120 detected with anti-TRP120 (top panel) and anti-Ub (bottom panel).

**Figure 3 pathogens-10-00962-f003:**
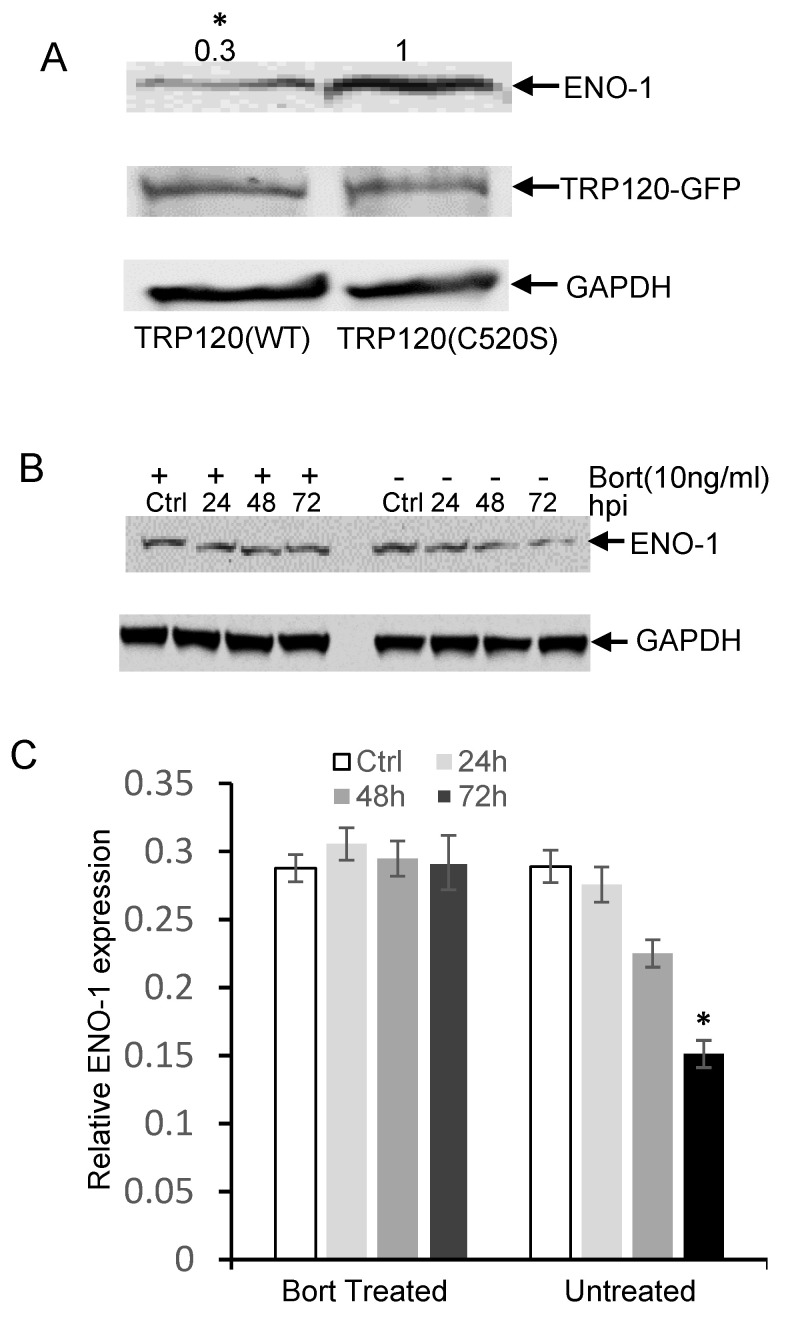
*E. chaffeensis* TRP120 HECT E3 Ub ligase activity mediates the degradation of ENO-1 by proteasome. (**A**) HeLa cells were transfected with TRP120 WT and HECT E3 ligase catalytic mutants (TRP120-C520S). Increased degradation of endogenous ENO-1 was detected in TRP120-WT compared to TRP120-C520S mutants lacking Ub ligase function. Densitometry fold change values of ENO-1 for TRP120-WT sample group are labeled above the band (TRP120-WT:0.3) compared with levels of ENO-1 for TRP120-C520S-mutant group (TRP120-C520S:1.0) (Student’s test; *, *p* < 0.05). (**B**) Western immunoblots demonstrating the effect of bortezomib (bort, 26S proteasome inhibitor) on ENO-1 during infection. Whole cell lysates were obtained from both bort-treated and untreated groups infected with cell-free *E. chaffeensis* at 24, 48 and 72 hpi and uninfected controls. ENO-1 levels remained unchanged during infection in the bortezomib-treated group; however, there was a temporal reduction in ENO-1 levels in the untreated group, demonstrating ENO-1 proteasomal degradation during infection. (**C**) Densitometry of Western immunoblots (**A**) performed using image J. Statistical analysis was carried out by comparing data from infected and control cells. Significance (*, *p* < 0.05) was determined for ENO-1 level at 72 hpi in the untreated group compared to uninfected control.

**Figure 4 pathogens-10-00962-f004:**
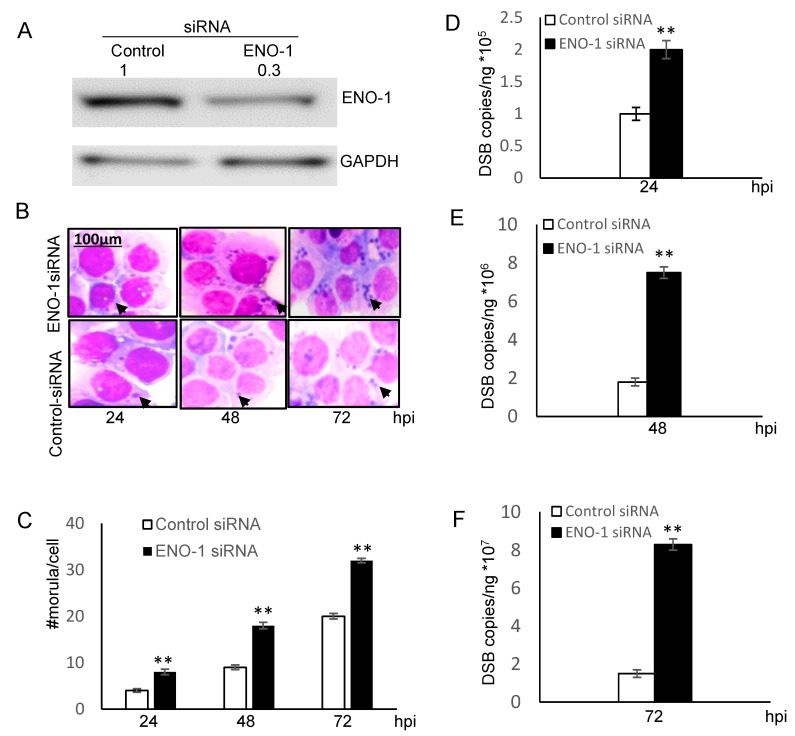
Knockdown of ENO-1 promotes ehrlichial infection. (**A**) Western blots were performed to determine knockdown efficiency. Densitometry fold change values of ENO-1 knockdown sample group are labeled above the band (ENO-1 siRNA:0.3) compared to the control siRNA group (Control siRNA:1.0). (**B**) Bright-field images (magnification, 1000×) of Diff-Quik-stained samples collected at 24, 48 and 72 hpi demonstrate increased number of ehrlichial inclusions/cell following ENO-1 knockdown. (**C**) Ehrlichial morulae/cell were determined by counting 20 cells/condition. ENO-1 knockdown infected samples were compared to control scrambled siRNA-transfected and infected cells to test significance (Student’s test; **, *p* < 0.01). (**D**–**F**) THP-1 cells were transfected with target or control siRNA and then infected by *E. chaffeensis*. Infection status was determined by *dsb* qPCR at 24, 48 and 72 hpi compared to control scrambled siRNA-transfected cells and normalized to host *GADPH* gene. Results shown are mean (± standard deviation) from three independent experiments (Student’s test; **, *p* < 0.01).

**Figure 5 pathogens-10-00962-f005:**
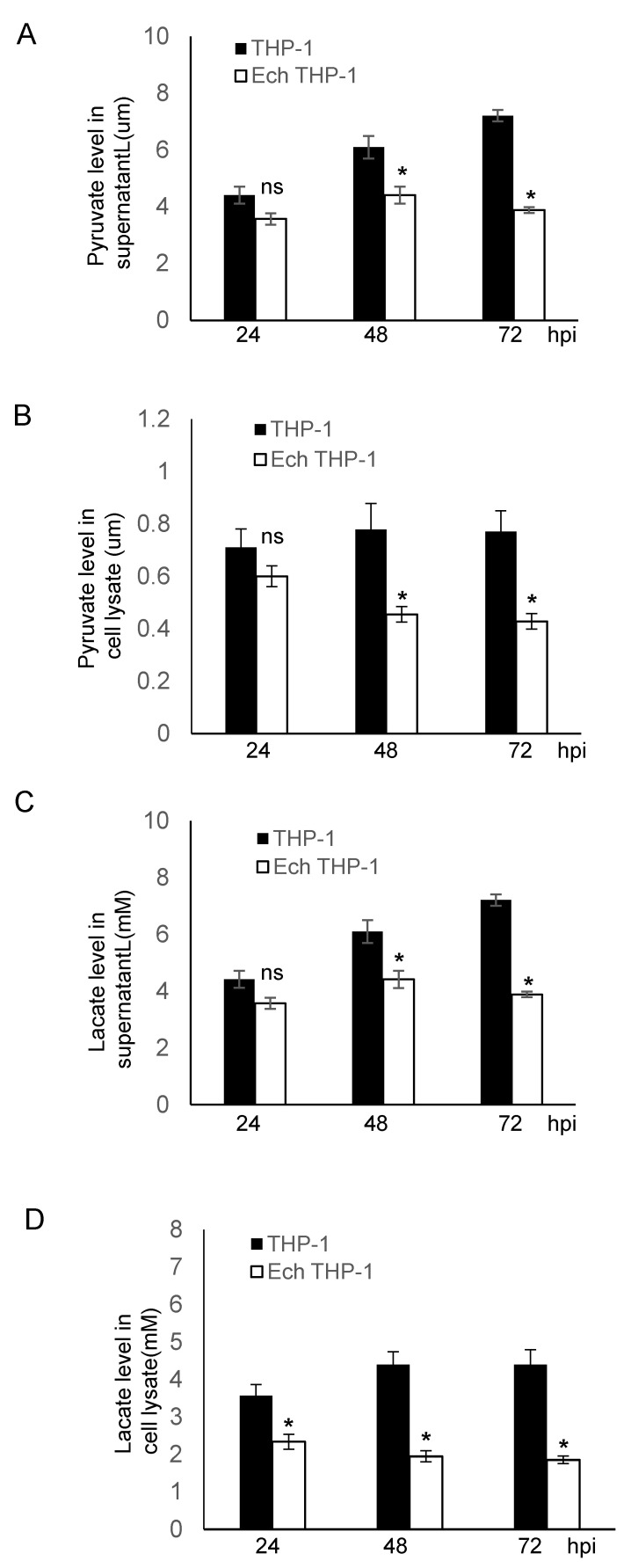
*E. chaffeensis* infection alters host pyruvate and lactate metabolism. (**A**,**B**) Pyruvate concentration in supernatant and cell lysate was determined after infection at different time points for 24, 48 and 72 hpi. The concentration of pyruvate in both supernatant and cell lysate was significantly lower than in infected cells. (Student’s test; *, *p* < 0.05; N.S., not significant). (**C**,**D**) Lactate concentration in supernatant and cell lysate was determined after infection at different time points for 24, 48 and 72 hpi. The concentration of lactate in both supernatant and cell lysate was significantly lower than in infected cells. (Student’s test; *, *p* < 0.05).

**Figure 6 pathogens-10-00962-f006:**
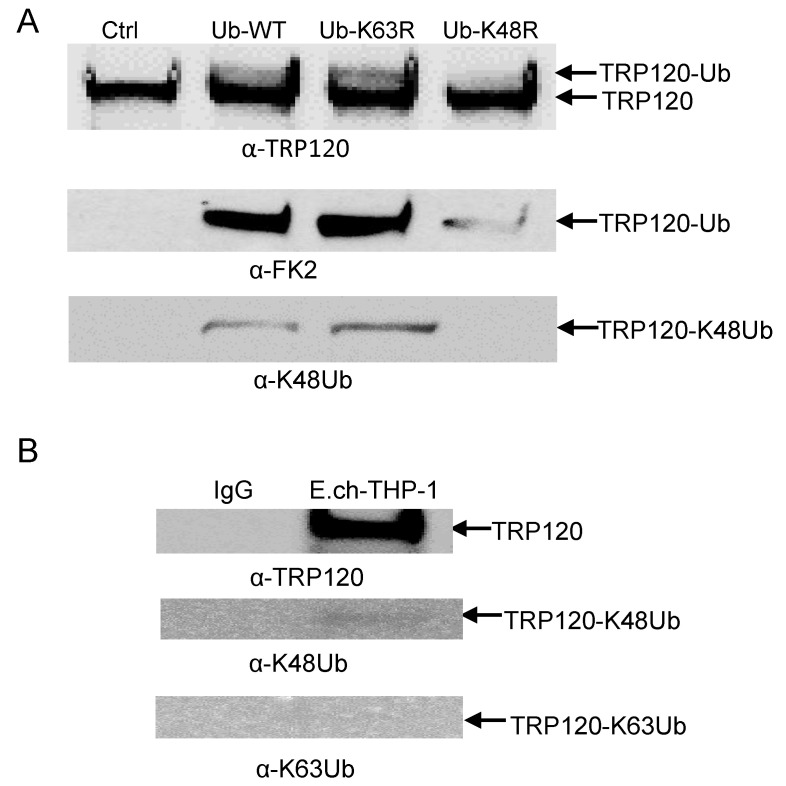
TRP120 ubiquitin linkage analysis in vitro and in vivo. (**A**) TRP120 in vitro ubiquitination of recombinant TRP120 in the presence of E1, UbcH5b E2 enzymes and K48R and K63R ubiquitin mutants. Immunoblot analysis of assay products with anti-TRP120, anti-FK2, anti-K48 and anti-K63 antibodies demonstrating TRP120 K48 ubiquitination. (**B**) Immunoprecipitation of *E. chaffeensis*-infected THP-1 cells. Immunoblot analysis of native TRP120 immunoprecipitated from *E. chaffeensis-*infected cells demonstrating ubiquitin linkage with K48 and not K63.

**Figure 7 pathogens-10-00962-f007:**
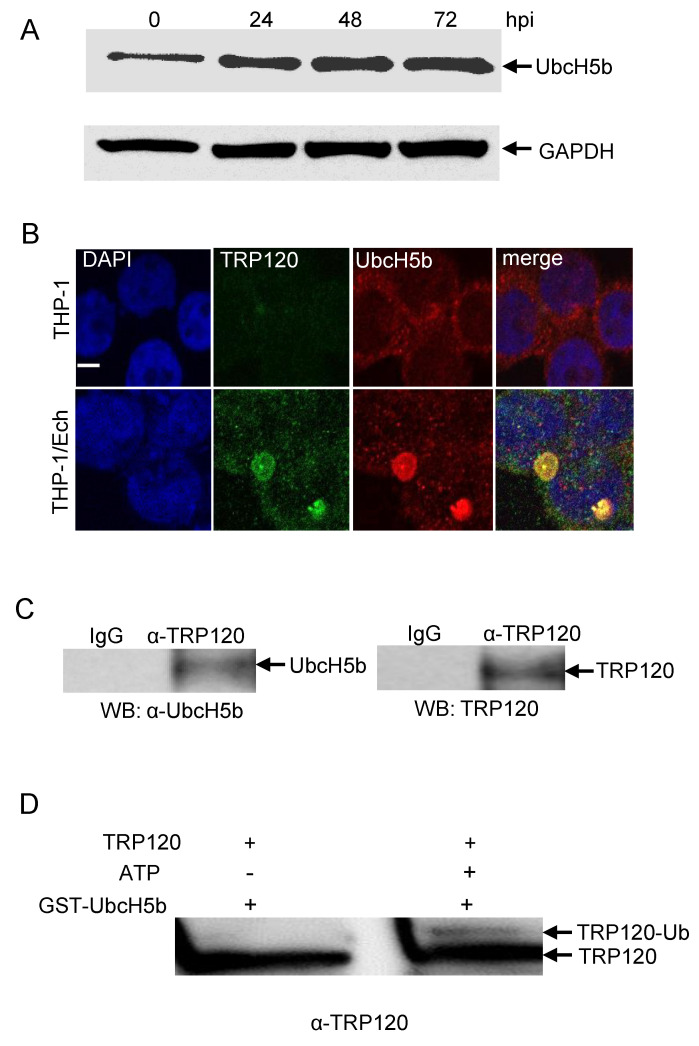
UbcH5b colocalizes with *E. chaffeensis* inclusions and interacts with TRP120. (**A**) Temporal immunoblot analysis of UbcH5b protein levels in the uninfected and *E. chaffeensis-*infected THP-1 cells. (**B**) Immunofluorescence microscopy visualization shows that UbcH5b colocalized with TRP120-expressing *E. chaffeensis* morulae. THP-1 cells were probed with DAPI (DNA; blue), anti-TRP120 (green) and anti-UbcH5b (red). *E. chaffeensis-*infected THP-1 cells were probed (72 hpi) with DAPI (DNA; blue), anti-TRP120 (green) and anti-UbcH5b (red) (bottom panel). Bar, 10 µm. (**C**) Western immunoblot detection of TRP120 and UbcH5b from immunoprecipitated samples prepared from *E. chaffeensis*-infected cells using TRP120- and UbcH5b-specific antibodies. (**D**) In vitro binding of GST-UbcH5b with unmodified TRP120 (without ATP) and ubiquitinated TRP120 (with ATP). Western immunoblot detection of TRP120 using anti-TRP120 antibody.

## Data Availability

Not applicable.

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
