# Peer review of "Alpha Enolase 1 Ubiquitination and Degradation Mediated by Ehrlichia chaffeensis TRP120 Disrupts Glycolytic Flux and Promotes Infection"

_pathogens, 2021, doi:10.3390/pathogens10080962_

Round 1

Reviewer 1 Report

The manuscript written by Zhu and McBride focused on the role of E. Chaffeensis in redirecting cellular metabolism through ENO-1 degradation by the ubiquitin-proteasome pathway during infection.

The only observation concerns the final discussion in which the authors should better clarify how the results described in the paper can be used for the development of new therapeutic strategies and the advantages associated with them.

Author Response

The manuscript written by Zhu and McBride focused on the role of E. chaffeensis in redirecting cellular metabolism through ENO-1 degradation by the ubiquitin-proteasome pathway during infection. 

The only observation concerns the final discussion in which the authors should better clarify how the results described in the paper can be used for the development of new therapeutic strategies and the advantages associated with them. 

Response: It is well known that ENO-1 has glycolytic and non-glycolytic functions and participates in a high number of cellular processes, suggesting that ENO-1 targeted therapeutic approaches may be considered. Future studies to clarify the role of functions of ENO-1 and its metabolites during infection will help better understand how such therapies could be developed. We have acknowledged this potential in the discussion  section.  

Reviewer 2 Report

In this study, Zhu and McBride investigated the interactions of TRP120, a multifunction Ehrlichia chaffeensis effector protein, for its role in altering alpha enolase (ENO-1) levels in infected host cells.  Multiple experiments were carried out to demonstrate downregulation of ENO-1 resulting from E. chaffeensis infection and is mediated by associating with TRP120 leading to degradation by the ubiquitin-proteasome pathway.   The data are strong and add more knowledge regarding the functional roles of the bacterial TRP120 for host-pathogen interactions to promote the pathogen’s continued replication in the hostile host cell environment.  The following are few points which need to be addressed in the revision.

1) Introduction (3rd paragraph): ENO-1 is not considered a key glycolytic enzyme. Its role in glycolysis is to remove water molecule from 2-phosphoglycerate, thus converting it to phosphoenolpyruvate.  The authors state that it is an important step in ATP generation which is also not correct.  However, as cited references in the discussion section, ENO-1 has additional functions beyond the glycolysis.  The authors should revise the introduction to accurately reflect enolase’s activities. 

2) Downregulation of ENO-1 probably cause the host cell to produce fewer energy due to downregulation of glycolysis which may in turn impact the overall ATP generation also from the citric acid cycle. How will this be a beneficial for the E. chaffeensis growth?  Additional discussion will be valuable to address this point.

3) It is highly likely that ENO-1 knockdown by siRNA may adversely impact the growth of host cells. Did the authors check the impact of siRNA knockdown of ENO-1 on the THP-1 cell replication?   

4) Lactate is produced from glycolysis only under anaerobic conditions, although low level lactate is produced in a cell for the purpose of lactate shuttle. During normal physiological conditions, pyruvate transfers to mitochondria for further degradation to generate ATP through citric acid cycle.  Thus, it is less likely that ENO-1 downregulation has a direct role in lactate reduction.

5) The following statements (in the discussion section section) are not scientifically supported with the data presented in the manuscript.

“These results suggest that ENO-1 is an essential enzyme molecule that regulates pyruvate and lactate production during infection, and elimination of ENO-1 slows the rate of production of pyruvate and lactate. These results also suggest that the degradation of ENO-1 by TRP120 may be associated with metabolic reprogramming of host cells to promote infection.” 

Author Response

Reviewer 2 

In this study, Zhu and McBride investigated the interactions of TRP120, a multifunction Ehrlichia chaffeensis effector protein, for its role in altering alpha enolase (ENO-1) levels in infected host cells.  Multiple experiments were carried out to demonstrate downregulation of ENO-1 resulting from E. chaffeensis infection and is mediated by associating with TRP120 leading to degradation by the ubiquitin-proteasome pathway. The data are strong and add more knowledge regarding the functional roles of the bacterial TRP120 for host-pathogen interactions to promote the pathogen’s continued replication in the hostile host cell environment. The following are few points which need to be addressed in the revision. 

1) Introduction (3rd paragraph): ENO-1 is not considered a key glycolytic enzyme. Its role in glycolysis is to remove water molecule from 2-phosphoglycerate, thus converting it to phosphoenolpyruvate.  The authors state that it is an important step in ATP generation which is also not correct. However, as cited references in the discussion section, ENO-1 has additional functions beyond the glycolysis.  The authors should revise the introduction to accurately reflect enolase’s activities.  

Response: ENO1 is a glycolytic enzyme involved in glycolysis to catalyze the dehydration of 2-phosphoglycerate to phosphoenolpyruvate in the catabolic glycolytic pathway.  We agree and think ENO-1 is not a key glycolytic enzyme but is an important glycolytic enzyme. Moreover, conversion of 2-phosphoglycerate to phosphoenolpyruvate mediated by enolase does not produce ATP directly, however, among the steps involved in ATP generation through substrate-level phosphorylation, ENO-1 is responsible for the ATP-generated conversion of 2-phosphoglycerate to phosphoenolpyruvate during glycolysis. We have provided a new information and reference citations regarding these ENO-1 activities  in introduction section  

2) Downregulation of ENO-1 probably cause the host cell to produce fewer energy due to downregulation of glycolysis which may in turn impact the overall ATP generation also from the citric acid cycle. How will this be a beneficial for the E. chaffeensis growth?  Additional discussion will be valuable to address this point. 

Response: Many bacteria and viruses are known to hijack cellular metabolism for infection. They actively reprogram host cell metabolism to redirect glycolysis and mitochondrial TCA cycle intermediates towards the biosynthesis of lipid droplets, fatty acids, amino acids and nucleotides that are required for pathogen nutritional and survival needs. On the other hand, host cell metabolic switches and pathways control the duration and intensity of innate or adaptive immune activation and further enhancing infection. In either case, the end result is generally a reduction in the tricaboxylic acid (TCA) cycle and an induction of aerobic glycolysis (also called the Warburg effect), in which pyruvate generated from glycolysis is converted to lactate in the cytoplasm, generating two ATP molecules. It has been shown that disruption of the TCA cycle increases the ability of Salmonella Typhimurium to survive within resting and activated murine macrophages. Macrophages infected with M. tuberculosis exhibit reduced levels of TCA cycle and induces a similar shift towards aerobic glycolysis, Inhibition of this metabolic switch leads to reduced production of cytokine and enhanced growth of intracellular bacteria. The present study shows that both pyruvate and lactate levels are significantly reduced during infection and suggests that the degradation of ENO-1 by TRP120 may has a direct role in pyruvate reduction, resulting in a reduced TCA cycle and amino acid consumption, and altered mitochondrial localization. On the other hand, it is not clear if E. chaffeensis infection induces aerobic glycolysis during infection for its replication or inhibits this metabolic switch to dampen the host immune response (i.e. production of cytokines). Future research will further elucidate the role of TRP120-mediated ubiquitination and degradation of ENO-1 in promoting E. chaffeensis  infection. We have expanded the discussions to address this concern.   

3) It is highly likely that ENO-1 knockdown by siRNA may adversely impact the growth of host cells. Did the authors check the impact of siRNA knockdown of ENO-1 on the THP-1 cell replication?    

Response: In this study, we did not check the impact of siRNA knockdown of ENO-1 on the THP-1 cell replication. Future research will address this point and further elucidate the role of TRP120-mediated ubiquitination and degradation of ENO-1 in promoting E. chaffeensis infection.     

4) Lactate is produced from glycolysis only under anaerobic conditions, although low level lactate is produced in a cell for the purpose of lactate shuttle. During normal physiological conditions, pyruvate transfers to mitochondria for further degradation to generate ATP through citric acid cycle. Thus, it is less likely that ENO-1 downregulation has a direct role in lactate reduction. 

Response: We appreciate this important point. Under normal conditions in most cells, pyruvate is shuttled into mitochondria, where it is oxidized via the TCA cycle, eventually generating ATP which to promote cell growth. Conversely, once inside mitochondria, mitochondria LDH would catalyze the conversion of lactate back to pyruvate, which would be oxidized through the PDH reaction to acetylCoA which to induce cell growth and proliferation. Therefore, it is less likely that downregulation of ENO-1 has a direct role in lactate reduction. However, whether ENO-1 downregulation by TRP120 has a direct role in lactate production through an induction of aerobic glycolysis is less clear. Therefore, our future research will investigate if E. caffeensis infection could induce aerobic glycolysis and further demonstrate that ENO-1 downregulation by TRP120 has a direct role or not in lactate reduction. We have addressed this point in the revised discussion. 

5) The following statements (in the discussion section section) are not scientifically supported with the data presented in the manuscript. 

“These results suggest that ENO-1 is an essential enzyme molecule that regulates pyruvate and lactate production during infection, and elimination of ENO-1 slows the rate of production of pyruvate and lactate. These results also suggest that the degradation of ENO-1 by TRP120 may be associated with metabolic reprogramming of host cells to promote infection.”  

Response: These statements are found in the results section (ENO-1 knockdown alters host cell metabolism) based on the following experiments. Additional discussion has been added to address if ENO-1 downregulation by TRP120 has a direct role in lactate reduction.  

To further demonstrate that the decrease in pyruvate and lactate production was due to the degradation of ENO-1, we compared changes of pyruvate and lactate production before and after ENO-1 knockdown. After silencing of ENO-1, the extracellular and intracellular levels of pyruvate and lactate were decreased in uninfected cells (Fig S1 A, C, E and G). In addition, the levels of pyruvate and lactate were significantly decreased after ENO-1 knockdown during infection (Fig S1 B, D, F and H). To determine if other metabolic enzymes such as pyruvate dehydrogenase, which is involved in the conversion of pyruvate to lactate, we investigated pyruvate dehydrogenase levels and found that they were similar in uninfected and infected cells (Fig. S2). These results suggest that ENO-1 is an essential for regulating pyruvate and lactate production during infection, and elimination of ENO-1 slows the rate of production of pyruvate and lactate. These results also suggest that the degradation of ENO-1 by TRP120 is associated with metabolic reprogramming of host cells to promote infection.